

# Impact of climate forecasts on the microbial quality of a drinking
# water source in Norway using hydrodynamic modelling
Hadi Mohammed[1], Andreas Longva[1], Razak Seidu[1]
[1]Water and Environmental Engineering Group, Institute for Marine Operations and Civil Engineering,
Norwegian University of Science and Technology (NTNU) in Ålesund, Larsgårdsvegen 2, 6009
Ålesund, Norway
*Corresponding author:* Hadi Mohammed (hadi.mohammed@tnu.no)
**Abstract**. This study develops hydrodynamic and water quality models for long-term prediction of *E. coli*
concentrations at the raw water intake point of lake Brusdalsvatnet in Norway. The study is based on previously
observed concentrations of *E. coli* in the tributaries of the lake and local projections of precipitation and air temperature
in the region. The results indicate a gradual rise in the temperature of water at the intake point from the base year
(2017) through to year 2075. Shorter spring circulation and longer autumn circulation periods are expected in the lake
in future. Concentrations of *E. coli* at the intake point of the lake are expected to marginally increase in future. By the
year 2075, the models predict a 3 fold and 2 fold increase in *E. coli* concentrations respectively for the spring and
autumn seasons compared to current levels. The results is expected to provide the water supply system managers of
Ålesund with the information necessary for long term planning and decisions in the protection of the drinking water
source. The method used here can also be applied to similar water supply systems for developing effective risk
management strategies for recent and future scenarios.
**Keywords:** Climate change, *E. coli*, hydrodynamic modelling, lake circulation periods, precipitation,
temperature.

## 1 Introduction
The link between extreme weather events and waterborne disease outbreaks is well established in the literature (Patz
& Hahn 2013; Smith *et al*. 2014; Tornevi *et al*. 2014; Levy *et al*. 2016). With the imminent threat of changing climate
variables on the quality of freshwater resources, water treatment plants that are heavily dependent on surface water
bodies are particularly vulnerable. Microbial deterioration of surface water sources due to the extreme precipitation;
and the resulting impact on the integrity of water treatment plants and disease outbreaks has been widely reported
(Soh *et al*. 2008; Drayna *et al*. 2010; Leppi *et al*. 2012; Cann *et al*. 2013; Guzman Herrador *et al*. 2016; Jagai *et al*.
2015; Bezirtzoglou *et al*. 2011; Bush *et al*. 2014; Eisenberg *et al*. 2013; Khan *et al*. 2015, Barry *et al*. 2016; De Roos
*et al*. 2017). Furthermore, increasing concentrations of natural organic matter in surface water sources due to changes
in precipitation patterns and catchment attributes (Aryal *et al*. 2016) may challenge the efficacy of water treatment
processes, enhance the formation of disinfection byproducts and regrowth of bacteria in the water distribution network,
and result in waterborne disease outbreaks (Hurst *et al*. 2004; Bull *et al*. 2011; Wang *et al*. 2016; Abokifa *et al*. 2016).
The impact of these extreme events on water supply systems is likely to be more pronounced in temperate countries
such as Norway, where seasonal variations and increases in temperature and precipitation are expected in the future.
According to the Norwegian green paper on climate change adaptation, the mean annual temperature in Norway is
expected to increase from 2.3 °C to 4.6 °C by 2100, with the highest and least increases expected in the winter and
summer months respectively. During the same period, annual precipitation is expected to increase from 5% to 30%
with major seasonal variations and increased frequency of torrential rains (Ministry of the Environment 2010). These
future changes in precipitation events and temperature will lead to significant changes in water quality parameters
including pathogens (Delpla *et al*. 2009). A study on the microbial quality of Norwegian surface water bodies showed
an association between rainfall and increased loads of faecal indicator organisms into surface waters (Tryland *et al*.



2011). A recent study has also shown significant association between microbial organisms in Norwegian raw water sources and changes in land use, and rainfall in the catchment (Johannessen *et al*. 2015).

Apart from rainfall, water temperature variations have been shown to affect the growth and survival dynamics of microbial organisms in raw water sources (Harvell *et al*. 2002; Vital *et al*. 2012; Pachepsky *et al*. 2014; Abia *et al*. 2016). Variations in water temperature, which is controlled by factors such as air temperature, cloud cover, solar radiation and other geomorphometric factors (Oswald & Rouse 2004; Sharma *et al*. 2015), affects the hydrodynamic distribution of microorganisms through increased stratification (Oswald & Rouse 2004; Sahoo *et al*. 2011; Thorne & Fenner 2011). In addition, the onset of heavy rains causes destratification, altering the movement of microbial organisms-bearing particles within the waterbody (Brookes *et al*. 2005). Short term and long-term stratification and destratification mainly resulting from temperature changes result in water quality deterioration (Lawson & Anderson 2007; Shade *et al*. 2011; Comeau *et al*. 2012). Accordingly, the development of resilient and adaptable management strategies necessary for the provision of safe drinking water in Norway require quantitative estimation of potential impact of local projections of weather parameters such as temperature and precipitation on the quality of raw water sources.

There is increasing reliance on models and forecasts for planning and decision making for effective management of drinking water facilities (Refsgaard & Henriksen 2004; Wool *et al*. 2003; McIntyre & Wheater 2004). Among the variety of models, properly calibrated hydrodynamic and water quality models provide reliable means of tracking primary sources of microbial contamination in drinking water sources (Hoyer *et al*. 2015; McCarthy *et al*. 2016) and recreational water (Zhu *et al*. 2011). In addition, these models can describe the transport of contaminants within watershed and their fate once in the waterbody (Guber *et al*. 2014; de Brauwere *et al*. 2014; Liu & Chan 2015; Sokolova *et al*. 2015). When properly calibrated, hydrodynamic models can provide reliable information about the sources of fecal indicator bacteria within catchment of a water source as well as identifying which source has the potential of posing the greatest threat to the microbial quality of drinking water source at the intake point. For effective planning of measures to mitigate potential health risks associated with microbial contamination of raw water sources, an assessment of potential levels of fecal indicator organisms such as *E. coli* in various sections of the waterbody at a particular time is imperative.

The overall aim of this study was to develop a hydrodynamic model to assess the impact of climate change on the microbial quality of the raw water source of a water treatment plant in Norway. The specific objectives were to assess model the variations in the *E.coli* concentration at the raw intake point of the water treatment plant, with respect to climate changes in climatic variables in 2045 and 2075. Developing a climate-driven microbial quality hydrodynamic model will not only provide insight into potential effects of climate change on the microbial quality of raw water, but also help managers of the water treatment plants in adequately planning long-term mitigation strategies necessary for the provision of safe drinking water to the public. Further, as water treatment plants are usually designed and built with a long-life span ranging from 25 – 30 years, understanding climate impacts are critical to developing appropriate management strategies. Similar water treatment plants to assess the impact of climate change on their drinking water supply systems may therefore adopt the approach used in this study.

## 2 Materials and methods

### 2.1 Study lake and catchment characteristics

The Brusdalsvatnet Lake, located in the West Coast Region of Møre and Romsdal region in Norway was used as a case in the development of the climate-driven microbial hydrodynamic model. The lake is the main water source of the Ålesund water treatment plant that supplies drinking water to about 50 000 inhabitants in the city of Ålesund and adjoining communities. The drinking water treatment plant draws 55,000 $m^3$ of water daily from the lake at the southwestern section of the lake at a depth of 35 m (Fig. 1). The deepest part of the Lake is approximately 99 m. The Lake has a surface area of about 7.3 $km^2$ with a mountainous and heavily forested catchment area of approximately 30 $km^2$, and is surrounded by few settlements mostly in the northwestern and southwestern parts. In addition to the numerous smaller streams surrounding the lake, are four major streams that drain into the lake (Fig. 1). These major





streams are Årsetelva, Vasstrandelva, Slettebakk and Brusdalen, with average annual flow rates of 0.15 m3/s and 0.17
m3/s, 0.08 m3/s, and 0.06 m3/s respectively. Majority of these smaller streams are either snowmelt or rainfall-induced,
and dry up in most parts of the year.
The lake drains into a much smaller lake called Lillevatnet, which is located at the Western end of the lake. Regular
rainfall in the lake catchment flushes loads of decayed organic materials from the forest catchment into the lake
through the streams. Wild animals and birds in the catchment have the potential of significantly contributing to the
microbial contamination of the lake mainly from their droppings, and this may include *E. coli* and other pathogens of
concern to human health. Within the populated areas surrounding the northwestern part of the lake, leakages and
seepages from household septic tanks also have the tendency of adding to the microbial loads of the lake, since most
of these houses are in close proximity to the lake. In addition, a major wastewater pipe traverses along the northwestern
end of the Lake. A previous study that analyzed water sample from streams across the lake revealed that samples
collected along the northwestern end of the lake contained higher concentrations of thermotolerant coliform bacteria
of up to $1.95 \times 10^4$ CFU/100 ml (Berg 2002). These high concentrations occurred at the populated areas within which
the sewage pipe traverses.

## 2.2 Hydrodynamic Modelling

The data used as inputs for the  hydrodynamic and water quality model included historical and projected
meteorological data, hydrological data (stream flow), geographical information system (GIS) data for the shape and
bathymetry of the lake, as well as historical and projected concentrations of *E. coli* in the streams.

### 2.2.1 Hydrological flows into the lake

Currently, the catchment of the lake is completely ungauged. Therefore, to efficiently account for inflows and
microbial discharges from the various streams surrounding the lake for use as inputs to the hydrodynamic model,
hydrological and water quality models were developed as described in a previous study (Mohammed *et al*. (submitted
manuscript 2018)). The hydrological and water quality models were based on sub-catchments using the soil and water
quality modeling tool (SWAT). The SWAT is a physically based and spatially distributed hydrological model used in
the simulation of water flow, sediments and contaminant transport within ungauged catchments (Arnold *et al*. 1998;
Arnold *et al*. 2012). Water inflows into the lake from the four major streams (Arsetelva Vasstrandelva, Slettebakk,
and Brusdalen) and their sub-catchments were targeted in the hydrological modeling. The models were developed
using hydrological parameter regionalization. That is, the model was initially developed from daily records of
precipitation and air temperature observed in the catchment of a nearby gauged lake (Engsetdalsvatnet: Lat. 62.53111,
Long. 6.64889) between 2010 and 2017, and the model parameters transferred to the catchment of our study area.
Hydrological model parameter regionalization offers an efficient means of estimating flows in ungauged catchments
from gauged ones that are in close proximity and share similar characteristics such as climate, topography, soil type
and land use (Bárdossy 2007). The models were subsequently validated with flows from the donor catchment
(Engsetdalsvatnet), which were scaled according to the catchment sizes of the rivers and streams modeled. For more
information and details on this approach and the results, readers are referred to (Mohammed *et al*. (submitted
manuscript 2018)).
Once the hydrological models were developed and validated, we used the parameters in combination with historical
observations of precipitation and air temperature adjusted from local climate projections for 2045 and 2075 to predict
flows in the major streams for these future years. For the smaller streams (S1-S4), historical and future flows were
estimated by calculating the difference between the inflows from the major streams and the sum of the outflow from
the lake and withdrawal from the water treatment plant. In addition, to account for discharge from areas that were
either not assessable for regular sampling (due to steep topography) or contain transient streams that only flow during
high precipitation periods, we created four additional discharge points during the implementation of the hydrodynamic
model. Therefore, the calculated flow difference between the inflow and outflow was distributed amongst the four



smaller streams (S1-S4) and the additional discharge points using their sub-catchment sizes as guides. Finally, a
hydrodynamic and water quality model for the drinking water source (Brusdalsvatnet Lake) was developed for 2017,
2045, and 2075 using the hydrological model results from the SWAT model and the estimated flows for the smaller
streams as inputs. Table 1 shows the average historical and future flows from the SWAT model for the major streams
and the calculated flows for the smaller streams.

**2.2.2 Microbial discharge into the lake**

Microbial discharge into the lake was accounted for by surface runoffs and direct discharges from the streams. We
carried out biweekly sampling and analysis for *E. coli* in the eight streams (Fig. 1) between March 2017 and February
2018, and these were used as inputs to the hydrodynamic model base year (2017). To obtain corresponding
concentrations of *E. coli* in the streams for the future, we calibrated additional water quality models for the four major
streams as part of the rainfall-runoff models in SWAT. In the SWAT model, loading of *E. coli* in the catchment are
introduced into hydrological response units (HRUs) in the form of dry animal manure within the catchments. In
addition, mass transport and die-off/regrowth equations are used to model the discharge and die-off of *E.coli* in the
soil surface layer (top 10 mm) and in the streams (Neitsch *et al*. 2011). The SWAT models were validated with the
observed *E. coli* in the streams and the models were subsequently used to predict *E. coli* concentrations in 2045 and
2075 using adjusted catchment precipitation and air temperature for the future as inputs. Owing to the sizes of the
smaller streams (S1 – S4), it was not possible to implement hydrological models for them in SWAT, which requires
definition of distinct stream channels within a digital elevation model (DEM). Therefore, future concentrations in
these streams were assumed to remain the same.
Table 2 shows a summary of the average concentrations of *E. coli* in the streams for the sampling period as well as
the SWAT model predictions for 2045 and 2075. Since the water utility managers plan to maintain the current land
use configurations within the catchment area of the raw water source over the coming years, it is assumed that the
only factors that will significantly determine the microbial quality of the lake are rainfall and temperature. For each
of the additional points, the concentrations of *E. coli* in the closest sampled stream was assigned, under the assumption
that two discharge points close to each other share similar spatial characteristics and potential sources of faecal
indicator organisms.

**2.2.3 Meteorological data**

The meteorological data used in the hydrodynamic model development were obtained from the Norwegian
Meteorological Institute. The weather stations included Vigra (Lat. 62. 5617, Long. 6.115), Hildre (Lat. 62.6017,
Long. 6.3187), and Ålesund IV (Lat. 62.4703, Long. 6.2108). The data constituted hourly observations of air
temperature, pressure, relative humidity, wind speed, wind direction, and cloud cover over the study area for the base
year of 2017. In addition, the hydrodynamic modeling software is composed of a time-varying data generator tool,
which was used to calculate hourly rates of surface heat exchange from the weather variables. Subsequently, the tool
computes a continuous water temperature data by applying a simple water temperature model (ERM 2006):

$$D\frac{dT}{dt} = \frac{R_n}{\rho C_p} \tag{1}$$

where $D$ is mean depth of water column, $t$ is time, $\rho$ is the water density, $C_p$ is the specific heat capacity of water ,
and $R_n$, the net rate of surface heat exchange which is computed as:

$$R_n = R_s - R_{sr} + R_a - R_{ar} - R_b - R_e - R_c \tag{2}$$

where $R_s$ and $R_{sr}$ are transmitted and reflected shortwave solar radiation, $R_a$ and $R_{ar}$ are the respective longwave
atmospheric radiations, $R_b$ is back radiation, $R_e$ is the heat loss through evaporation, and $R_c$ is conducted heat.





For the future scenarios, the historical time series of temperature was adjusted using biase-corrected projections of air
temperature in the region. The method which is commonly used in hydrological climate impact assessments
(Teutschbein & Seibert 2012; Shrestha *et al*. 2017), involves transformation of historical time series of climate
variables with the ratio between mean future and historical climate projections. In this study, the Norwegian Water
Resources and Energy Directorate (NVE) based temperature projections on Representative Concentration Pathways
(RCP 8.5) climate models for the Møre and Romsdal region of Norway, where the study site is located, were used.
The RCP 8.5 projections is composed of results from ensembles of climate models (10 different models), which use
the period 1971 – 200 as the base year and predict climate change for up to 2100 as a moving average (40 - average).
Therefore, the median values of the model projections for 2045 and 2075 were used data used in this study. Since no
projections of the other weather variables (pressure, relative humidity, wind speed, wind direction, and cloud cover)
were available at the time of this study, we applied the historical values to the future hydrodynamic model scenarios.
Finally, GIS data for the lake shoreline and the bathymetry were used to define the boundaries of the lake for the
hydrodynamic computation.

**2.2.4 Implementing the hydrodynamic and water quality models**
The Generalized Environmental Modeling System for Surfacewaters GEMSS software (ERM 2006a, b) was used to
develop hydrodynamic and transport models from the GIS and ecological data. The theoretical basis of the system
computations are the longitudinal-vertical transport model (Buchak & Edinger 1984) developed from the horizontal
momentum balance, continuity equation, constituent transport and the equation of state. For the horizontal velocity
components $u$ and $v$ in the x and y - directions and the depth $z$ measured from the surface, the momentum balances
are:

$$\frac{\partial u}{\partial t} = g\frac{\partial z'}{\partial x} - \frac{g}{\rho}\int_{z'}^{z}\left(\frac{\partial \rho}{\partial x}\right)\partial z - \left(\frac{\partial uu}{\partial x} + \frac{\partial vu}{\partial y} + \frac{\partial wu}{\partial z}\right) + \left(\frac{\partial A_x}{\partial x}\left(\frac{\partial u}{\partial x}\right) + \frac{\partial A_y}{\partial y}\left(\frac{\partial u}{\partial y}\right) + \frac{\partial A_z}{\partial z}\left(\frac{\partial u}{\partial z}\right)\right) + fv - SM_x$$

203                                                                                                                                                      (3)

$$\frac{\partial v}{\partial t} = g\frac{\partial z'}{\partial y} - \frac{g}{\rho}\int_{z'}^{z}\left(\frac{\partial \rho}{\partial y}\right)\partial z - \left(\frac{\partial uv}{\partial x} + \frac{\partial vv}{\partial y} + \frac{\partial wv}{\partial z}\right) + \left(\frac{\partial A_x}{\partial x}\left(\frac{\partial v}{\partial x}\right) + \frac{\partial A_y}{\partial y}\left(\frac{\partial v}{\partial y}\right) + \frac{\partial A_z}{\partial z}\left(\frac{\partial v}{\partial z}\right)\right) - fu - SM_y$$

205                                                                                                                                                      (4)

where $z'$ is the elevation of the water surface, $fu$ and $fv$ are the Coriolis accelerations in the $x$ and $y$ directions, and
the terms $SM_x$ and $SM_y$ are the discharges into the Lake from the tributaries. The terms $A_x$, $A_y$ and $A_z$ are the
constituent dispersion coefficients. To compute the corresponding vertical component of the velocity ($w$), the local
continuity and the vertically integrated continuity for the surface elevation are:
$$\frac{\partial w}{\partial z} + \frac{\partial u}{\partial x} + \frac{\partial v}{\partial y} = 0$$                                                    (5)
$$\frac{\partial z'}{\partial t} + \int_{z}^{h}\frac{\partial u}{\partial x}dz + \int_{z}^{h}\frac{\partial v}{\partial y}dz = 0$$                          (6)

Transport of energy and constituents such as *E. coli* in the water is computed for each grid cell at each time step using
the equation:
$$\frac{\partial C_n}{\partial t} = -\left(\frac{\partial uC_n}{\partial x} + \frac{\partial vC_n}{\partial y} + \frac{\partial wC_n}{\partial z}\right) + \left(\frac{\partial D_x}{\partial x}\left(\frac{\partial C_n}{\partial x}\right) + \frac{\partial D_y}{\partial y}\left(\frac{\partial C_n}{\partial y}\right) + \frac{\partial D_z}{\partial z}\left(\frac{\partial C_n}{\partial z}\right)\right) + H_n$$





$\hfill$ (7)
where $C_n$ is the constituent with number $n$. The term $H_n$ in equation 8 accounts for all other sources and sinks of the
constituent. Finally, the equation of state, which relates the density of water to the constituents, is computed as
$$\rho = fn(C_1, C_2, C_3 \ldots, C_n) \hspace{2cm} (8)$$
where $f_n$ is the density function. The function used in this study is the one proposed by Gill (1982).
Using these equations, the system computes the concentration of *E. coli* in the lake as well as temperature from 3-D
time-varying flow fields and elevations for each computational cell of size (1 m x 1 m x 1m) along the horizontal and
vertical dimensions of the lake. In this study, an upwind first order scheme of constituent transport was used in a fully
explicit method such that all the terms that enter the computation of the constituent are derived from prevailing time
step. An in-depth numerical analysis of the computations that takes place in each grid cell and time step can be found
in Buchack & Edinger (1984). Further, the semi-implicit transport scheme is described in Smith (2006).

### 2.2.5 Mesh generation

The GEMMS software used in this study is integrated with a grid generation tool. This tool was used to generate
square grids of dimension 1m within the boundaries of the lake. Thus, the longitudinal and lateral dimensions of the
lake were divided into 100 and 25 cells respectively, with 94 vertical layers from the water surface to the bottom. Fig.
2 shows the generated grids for the surface of the lake. The model was developed to simulate the temperature and *E.*
*coli* transport in the lake for 2017. To validate the model, weekly measured water temperature and observed counts of
*E. coli* at the raw water intake point were compared with the outputs of the model. Subsequently, the adjusted air
temperature as well as the predicted flow and concentrations of *E. coli* in the streams for 2045 and 2075 were used as
inputs for simulating the future scenarios. Simulation for each year was performed from January to December with an
initial lake water temperature of 4 ºC, assumed from the value of 4.5 ºC measured at the treatment plant in January
2017. We applied a lower water temperature because the measured temperature at the plant may not represent actual
level at the raw water intake point of the treatment plant, which is 35 m below the lake surface. The *E. coli*
concentrations for each year was entered at the same time with a decay rate of 0.67 per day.  The output of the
simulations were in the form of time series, contours and profiles put together as access files and these were further
processed to generate desired figures to be analyzed.

### 3 Results

### 3.1 Hydrodynamic model validation

Fig. 3 shows a comparison of the hydrodynamic model outputs with measured temperature from the water treatment
plant in 2017. The raw water temperature values used to validate the model were measured in the treated water
reservoir whereas the model outputs represent values at the actual raw water withdrawal point of 35 m below the
surface. The simulated water temperature values were generally close to the measured values in winter (December -
February), Spring (March - May), as well as in autumn (September - November). However, the precision of the model
in predicting the raw water temperature during the summer months (June - August) was very low, with average
difference of 1.2 ºC. In addition, while the peak temperature of the raw water was measured in the first week of
November 2017, the model predicted a peak during the third week of the month. The peak temperature values were
similar nonetheless (measured: 7.11ºC, predicted: 7.22 ºC). The disparities in the model outputs and the measured
temperature particularly in the summer season may have resulted from changes in the water temperature as it travels
through the withdrawal pipes from the intake depth (35 m) and through the treatment processes in the water treatment
plant. Moreover, while the water temperature was measured on a weekly basis, the model calculated water temperature
at time intervals between 10 and 180 seconds. Accordingly, the model outputs may reflect the actual water temperature
at the 35 m depth. We further compared temperature profiles measured at six different days in the Lake in 2017 at
depth intervals of 5 m with profiles taken from the model outputs on those days as shown in Fig. 3 (a). Although




measured temperature profiles were not available for the summer months, it can be seen in the Fig. that the model
closely predicted the profiles.
As shown in Fig. 3 (c), the predicted *E. coli* concentrations were generally in agreement with the patterns of variations
in the observations at the raw water intake point. There were only two positive observations of  *E. coli* concentrations
at the intake point of the water utility in 2017; 2 CFU/100 ml in the first week of January and 3 CFU/100 ml in the
fourth week of December. The model however predicted the occurrence of low concentrations (< 1CFU/100 ml) in
late winter (February) and in the spring months, with up to 3 CFU/100 ml in the autumn. This indicate that water
circulation in the Lake during these two seasons were predicted by the model, as this may result in the occurrence of
the microorganisms at deeper parts of the Lake in comparison with periods of stratification in summer due to low
inflows.  Moreover, while the model predicts the concentrations without necessarily treating microorganisms as count
variables, analysis of microorganisms present in water only identifies colonies that are counted. Thus, the model can
predict lower concentrations that are not accounted for during the analysis. We further compared the model outputs
with the *E. coli* concentrations observed in the raw water in 2015 (Fig. 3 (c)). Interestingly, the model outputs appear
to agree more with this data set than the 2017 observations.

**3.2 Temperature and *E. coli* distribution in the Lake in 2017**
Fig. 4 shows the distribution of temperature and concentration of *E. coli* in the Lake in 2017 during the four major
seasons. Water circulation and vertical convective mixing mostly occurring during spring and autumn seasons
characterize the Lake. During the spring circulation period of 2017 (Fig. 4 (a1)), a nearly isothermal condition was
observed throughout the entire depth of the Lake in the western section where the raw water intake is located. In this
season, the water temperature ranged from 1 °C in the top 30 m of the western section of the Lake, to approximately
4 °C in the eastern part. This period of circulation can be associated with snowmelt in the catchment that lead to high
flows into the Lake. As shown in Fig. 4 (a2), this circulation resulted in dispersion of *E. coli* from the locations of
high contamination sources (Slettebakk and Brusdalen streams) in the eastern section of the lake. However,
concentration of *E. coli* at the raw water intake zone in the western section was low (< 1 CFU/100 ml). During this
period, not only are the concentrations in the inflow streams likely to be elevated, traces of microorganisms that
survived the freezing temperature in the ice cover are also released. The autumn circulation (Fig. 4 (a1)) is caused by
the onset of rainfall after summer, and resulted in higher temperature (~ 9°C). While *E. coli* concentrations at the 35
m depth was in excess of 5 CFU/100 ml, the concentrations at the intake zone were low ( < 3 CFU/100 ml) (Fig. 4
(C2)). Below this depth, the temperature was low and stratified. Ice cover that characterize the surface of the Lake in
winter leads to low water temperature (< 4 °C) throughout the Lake as shown in Fig. 4 (a1). The ice cover retains large
proportion of the inflow stream water and their contaminant loads, resulting in low concentrations of *E. coli* in the
Lake. Additionally, the liquid phase of the streams could be reduced during these months, thereby lowering their flow
levels and habitable organic materials.
Positive temperature gradient was noted on the surface from the shallow shoreline areas of the Lake to the central
region with higher magnitude of variation occurring in the summer months. The almost seven months of summer
season is demonstrated by higher temperature at the water surface reaching a maximum of approximately 17 °C  in
late July and early August (Fig. 4 (b1)). During this period, the deepest layers of the stratified Lake remained cooler
at temperatures of 4°C. The model also showed intense thermal stratification of the Lake during this period, with a
negative temperature gradient from the surface to the deeper layers. Due to the intense stratification in summer, very
low concentration of *E. coli* < 0.5 CFU/100 ml) occurs at the raw water intake zone (Fig. 4 (b2)), although the higher
concentrations reach deeper layers in the eastern section of the Lake where the inputs from the streams were high.
This can be caused by high concentrations in the streams, which often occur in summer. In addition, the overall time
series of the *E. coli* concentrations from the observations and the model outputs were lowest in summer.





### 3.3 Predicted temperature and *E. coli* in 2045 and 2075

The predicted water temperature and *E. coli* concentrations in the Lake are presented in Fig. 5. In this figure, temperature at the surface and raw water intake depth (35 m) in 2017 are compared with the predictions for 2045 and 2075 (Fig. 5 (a-c)). The model results indicate same startup time of spring circulation for all the projected years. Just as year 2017, spring circulation period for year 2045 starts from middle of March and ends in late April, while the autumn circulation starts in late November. However, spring circulation in 2075 is likely to be one week shorter, ending in the third week of April. In addition, autumn circulation in this year is shifted forward by a week, starting early December. This indicate that period of high raw water temperature may increase by 2075 due to the longer summer. Further, the intensity of spring circulation may increase in the future, as the deeper water temperature increasingly approach that at the surface. The implication is that the chances of contaminants at the water surface reaching the deeper layers will be high due to perfect mixing. The longer summer seasons expected in the future will result in higher raw water temperature, with surface temperature reaching 18 ℃ and 19 ℃ respectively in 2045 and 2075 (Fig.5 (d)). This will result in higher temperature at the raw water intake depth (Fig. 5 (e)) during the autumn seasons (from 7 ℃ in 2017 to 8.6 ℃ in 2075).

The potentially late start of circulation period in the autumn seasons in future has the possibility of overriding winter conditions since the autumn circulation may extend until the start of the proceeding spring circulation. Higher concentrations of *E. coli* may therefore occur at the intake depth of the lake throughout the autumn, winter and spring in 2045 and 2075. As shown in Fig. 5 (f), maximum concentration of *E. coli* at the raw water intake depth increases from < 1 CFU/100 ml in the spring of 2017 to 2 CFU/100 ml in the same season of 2075. Similarly, the concentration in autumn increases from a maximum of 3 CFU/100 ml in 2017 to > 5 CFU/100 ml in 2045 and 2075.

### 4 Discussion

The hydrodynamic model simulation showed the overall effect of the *E. coli* discharged from the streams on the *E.coli* level throughout the Lake. The key sources of *E. coli* load to the Lake were the major streams namely including Årsetelva, Vasstrandelva, Brusdalen and Slettebakk. This may be partly due to their higher flows compared to the smaller streams, potentially causing circulation that is more turbulent. Circulation occurring in the Lake in the spring and autumn increased the chances of *E. coli* reaching greater depths in the Lake. Moderate rainfall at the turn over period following the long summer season partly account for the sudden rise in the concentration of *E. coli* towards the end of November, since they favor the accumulation and transport of organic and inorganic matter into the Lake through elevated stream flows. This result is consistent with a related study that reported high concentrations of *E. coli* in a Lake in Sweden during the same period and lowest levels in summer (Sokolova *et al*. 2013). Further, temperature distribution in the Lake (Fig. 4) indicate that considerable amount of vertical mixing of the Lake water occurred during this period, thereby increasing the transport of the bacteria to the water intake point of 35 m below surface. Moreover, high velocity wind currents, which characterize this season, enhance the circulation of water in the lake and this could increase the likelihood of contaminants reaching the intake depth.

Despite the overall very low *E. coli* concentrations predicted at raw water intake zone in summer, the cross-sections indicate high concentrations potentially occurring at the same depth in the section of the Lake with the highest source (Slettebakk and Brusdalen streams to the eastern side) as shown in Fig. 4 B2. The high concentrations in that part may be a reflection of the high concentrations in the streams already observed in summer. Potential sources of *E. coli* such as wild animals and birds in the catchment of the Lake are more active in this season, and may have contributed to the observed concentrations in the streams as well as the output of the model in this study. Further, although the inactivation rates of microbial organisms in surface water generally occur faster with increasing temperature, this dependency can be affected by site specific conditions and can vary among different water sources (Blaustein *et al*. 2013; Pachepsky *et al*. 2014). It is therefore possible that typical surface water temperatures in summer in the study region create favorable conditions for the survival of *E. coli* in the streams. While high concentrations of *E. coli* in the streams may be associated with catchment precipitation through increased flows and high sediment loads in spring and autumn, low flows in summer could lead to shorter travel distance and longer settling time in the streams and these may affect the concentrations of microorganisms in surface water (Schijven *et al*. 2013).



Nonetheless, the time series indicated generally very low concentrations in the Lake during this period (Fig. 5 (F)).
This also agreed with the observation in 2015 and 2017. It has been reported that other factors including lower loading
of fecal materials into surface water occurring during the summer season as well as potentially less viability of fecal
indicator organisms at higher water temperatures may contribute to this observed trend (An *et al.* 2002). In addition,
increased solar radiation in summer is reported as an important contributor to the inactivation of indicator bacteria in
large freshwater bodies such as Lakes (Whitman *et al.* 2004; Liu *et al.* 2006). Moreover, the thermoclines in the Lake
during this season separates the epilimion from the hypolimnion, restricting water circulation and the spread of
contaminants in lakes (Boehrer & Schultze 2008).
The model results generally indicate a pattern of water temperature and *E. coli* in 2045 and 2075 similar to the base
year (2017). However, an increasing trend of water temperature were observed across all the seasons. Water
temperature in spring, summer, autumn and winter rises by an average of 0.43 ℃, 1.2 ℃, 1.34 ℃, 0.89 ℃ respectively
by 2075 relative to 2017. The concentrations of *E. coli* at the water intake point in future may remain at levels close
to the observed concentrations presently observed in summer. The concentrations in spring and autumn may however
be higher than present levels, with the possibility of higher concentrations in winter due to late start of the autumn
circulation in future. Thus, based on current projections of precipitation and air temperature in the study region, plans
about the management of the drinking water facility should take into account the possibility of higher *E. coli* levels
occurring in the water.
The results of this study provides useful assessments of the effect of climate change on the microbial quality of the
raw water source for the treatment plant. However, the extensive use of climate data introduces considerable
limitations in the use of the results, therefore management decisions that will be taken based on the results should
consider such limitations. Major sources of uncertainties include the historical observations of the weather variables
used in both the previous hydrological models and the hydrodynamic model, the climate projections, as well as the
model formulations and their calibrations in this study. While uncertainties in the predicted stream flow and *E. coli*
concentrations were accounted for in the previous hydrological model that provided additional inputs for this study,
further assumptions were made about the concentrations of *E. coli* in the unmonitored and transient tributaries of the
lake during the implementation of the hydrodynamic model. Thus, discharges from those sections could be higher in
the future, potentially affecting the concentrations that reach the raw water intake zone. In addition, the method applied
in this study only account for the status quo scenarios that assume all other things in the catchment of the lake
remaining the same in the future. Although the water treatment plant managers plan to maintain current regulations to
limit further development and recreational activities within the catchment, incidents such as extreme weather events,
combined sewer overflows, or bursting of sewer pipes can potentially lead to sudden increases in the concentrations
of microorganisms discharged into the lake. However, such scenarios have not been accounted for in the present study.

## 5 Conclusions

Potential impact of climate change projections on water temperature and *E. coli* concentrations in a raw water source
has been undertaken with a focus on the Brusdalsvatnet Lake in Ålesund, Norway using a 3D hydrodynamic and water
quality modelling approach. Reasonable accuracies were achieved in both the water temperature and *E. coli*
predictions in the base year (2017). The model results for the years 2045 and 2075 indicate a gradual rise in the
temperature at the water intake point of the lake from the base year levels. In addition, shorter spring circulation and
longer autumn circulation periods are expected in the lake in future. Under the current climate forecasts for the
catchment area of the Lake, the concentrations of *E. coli* in the Lake, particularly at the water intake point of the
treatment plant in the Ålesund water treatment plant is expected to marginally increase by 2075. The results is expected
to provide the water supply managers of the water utility with the  information necessary for long term planning and
decisions in the protection of the water source. Moreover, with high quality hydrological, water quality and climate
data in the catchment of drinking water sources, the approach applied in this study may be useful for developing
effective risk management strategies for recent and future scenarios.





## 6 Acknowledgements

The Research Council of Norway, under the project "Impact of climate change on the association between extreme weather events and waterborne illness", and the Ålesund water treatment plant, provided funding for this research. The authors are also grateful to the Norwegian Water Resources and Energy Directorate (NVE) and the Norwegian Meteorological Institute (MET Norway) for the provision of data. We further extend our gratitude to the Water Resource and Modeling Team of ERM Inc. for providing us with the license for the GEMSS software used in this research. Finally, we acknowledge the support and inputs provided by Bjørn Skulstad of the Ålesund Kommune.

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





TABLES

Table 1. Average historical and future flows in the streams

| Stream | Flow (Historical) m3/s | Flow (2045) m3/s | Flow (2075) m3/s |
|---|---|---|---|
| Arsetelva | 0.215 | 0.248 | 0.257 |
| Vasstrandelva | 0.273 | 0.246 | 0.252 |
| Slettebakk | 0.084 | 0.092 | 0.098 |
| Brusdalen | 0.044 | 0.041 | 0.043 |
| S1 | 0.028 | 0.028 | 0.028 |
| S2 | 0.021 | 0.021 | 0.021 |
| S3 | 0.021 | 0.021 | 0.021 |
| S4 | 0.023 | 0.023 | 0.023 |

























Table 2. Average concentrations of *E. coli* in the tributaries from the monitoring exercise in 2017-2018 and the SWAT
model-predicted concentrations in 2045 and 2075.

| Source | Average concentration of *E. coli* (CFU/100 ml) | | |
|---|---|---|---|
| | 2017 | 2045 | 2075 |
| Årsetelva | 26 | 11 | 12 |
| Vasstrandelva | 77 | 22 | 23 |
| Slettebakk | 18052 | 14554 | 14711 |
| Brusdalen | 45524 | 38462 | 37964 |
| Stream 1 | 39 | 39 | 39 |
| Stream 2 | 11 | 11 | 11 |
| Stream 3 | 210 | 210 | 210 |
| Stream 4 | 50 | 50 | 50 |








FIGURES

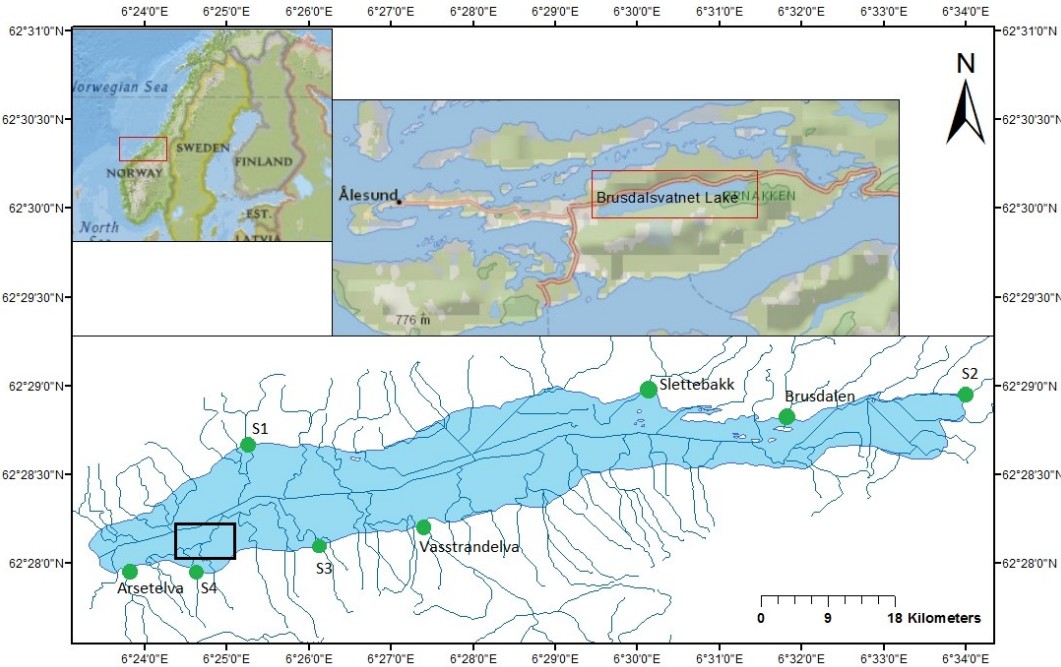

**Figure 1.** Map of Brusdalsvatnet Lake showing the locations of the various streams (green spots) and the raw water
intake zone of the water treatment plant (black rectangle). S1 – S4 are smaller streams.




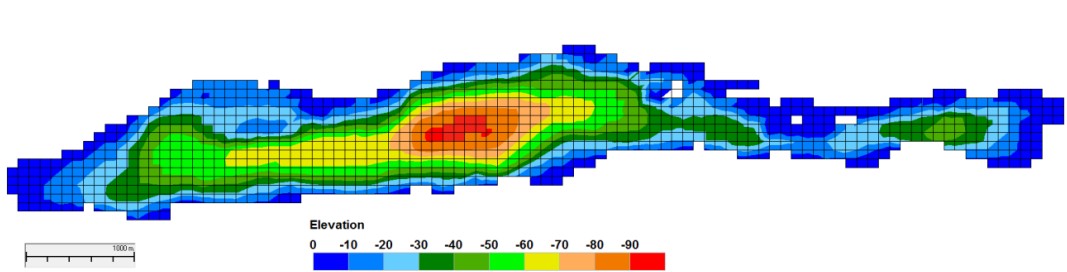

**Figure 2.** Computational mesh and bathymetry of Brusdalsvatnet Lake.






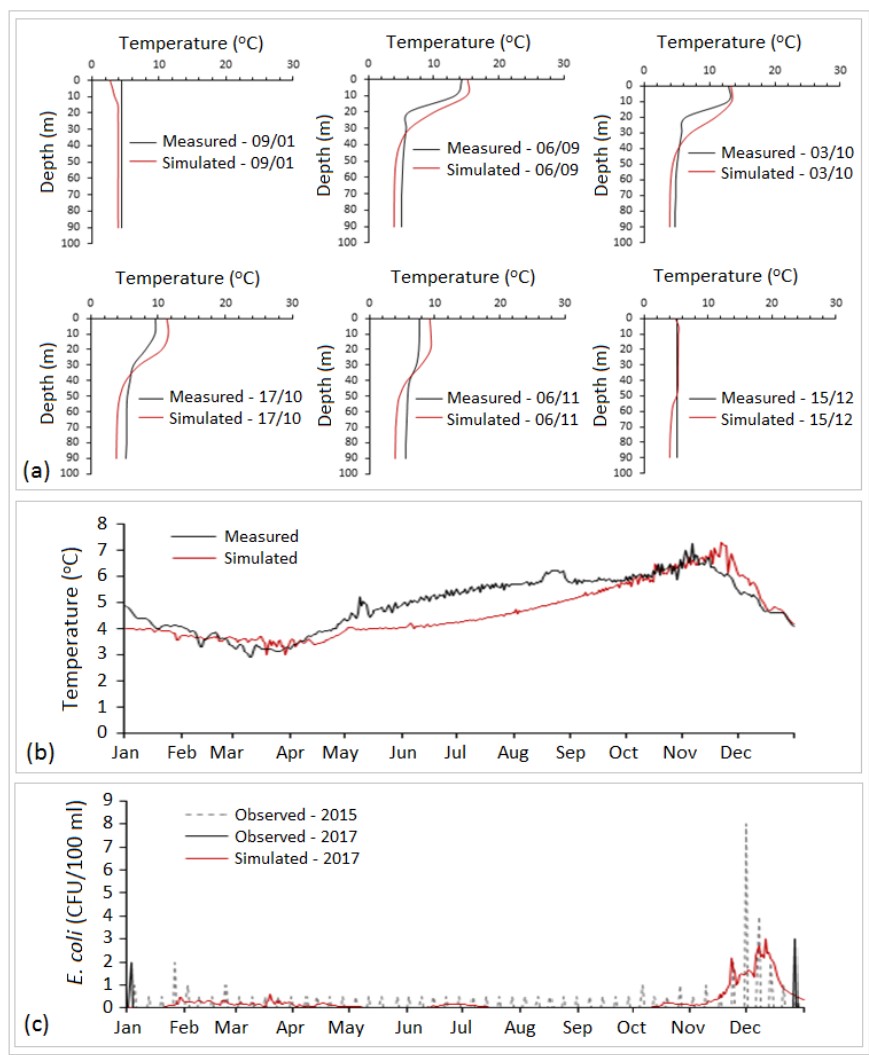

**Figure 3.** (a) Comparison of model outputs temperature profiles, (b) measured temperature at raw water intake point,
and (c) observed concentrations of *E. coli* in the raw water in 2015 and 2017




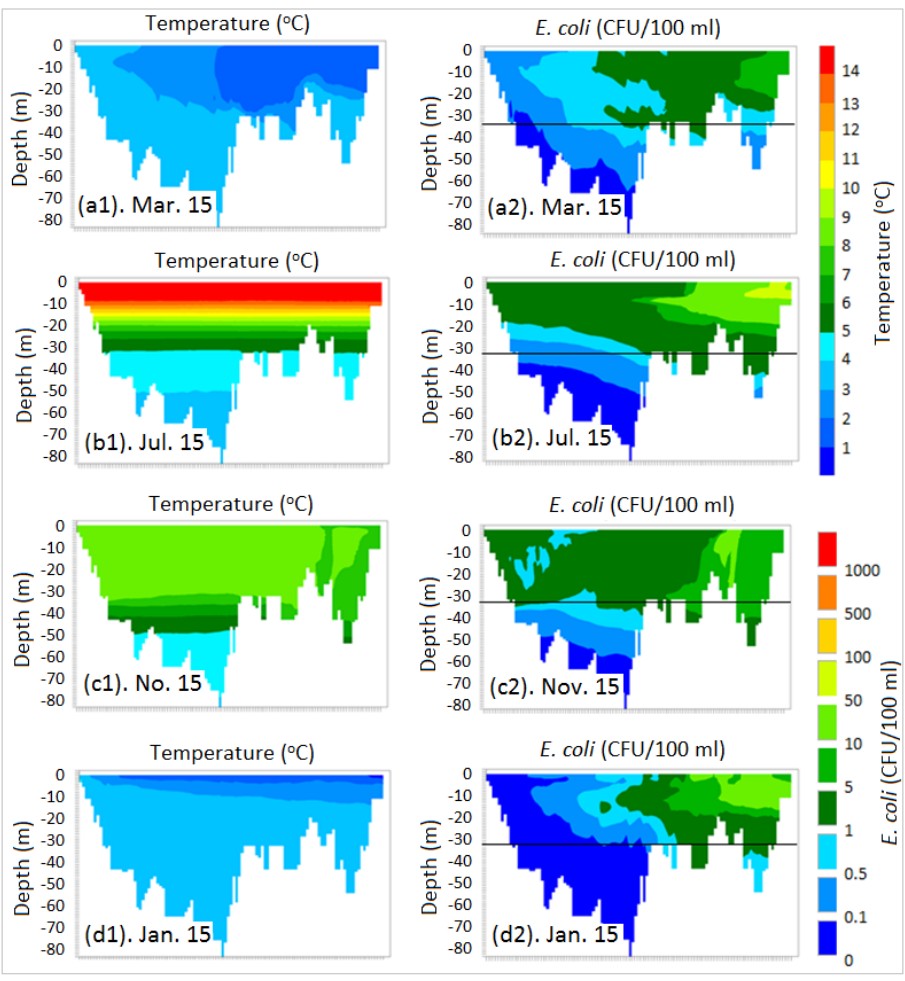

**Figure 4.** Cross-sections from the model output showing the distribution of temperature and *E. coli* in Brusdalsvatnet
Lake in spring (a1 and a2), summer (b1 and b2), autumn (c1 and c2) and winter (d1 and d2). The black lines indicate
the raw water intake depth.




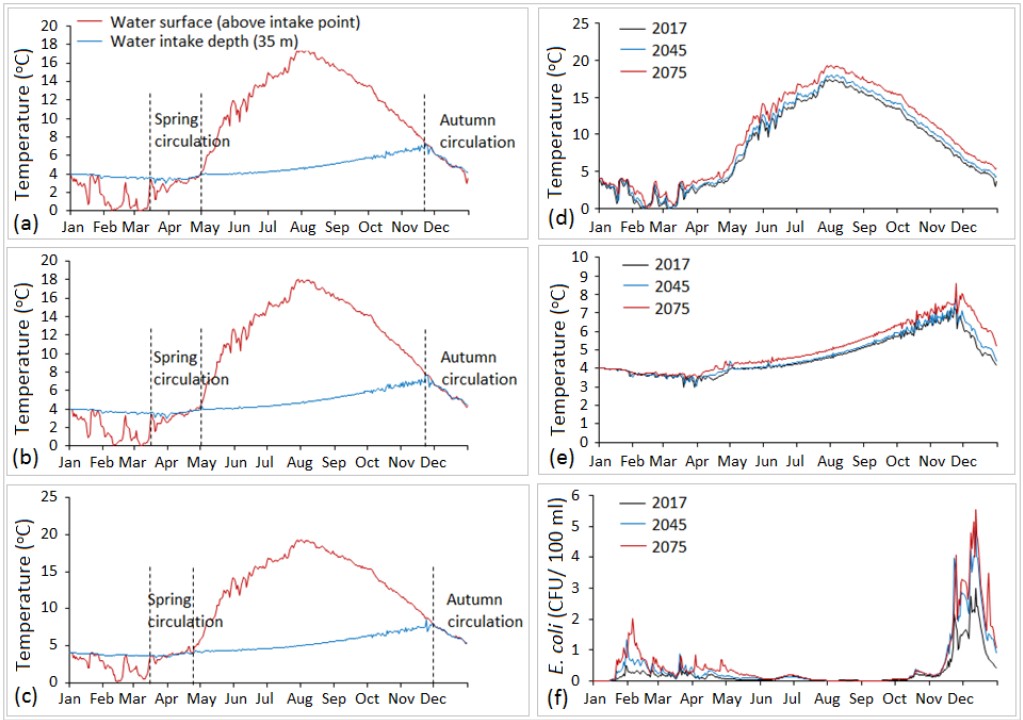

**Figure 5.** Comparison of temperature at the lake surface and raw water intake point for 2017 (a), 2045 (b), and 2075
(c). Increases in water surface temperature (d), intake temperature (e), and *E. coli* (f) in these years are also shown.