# Peer review of "Impact of climate forecasts on the microbial quality of a drinking water source in Norway using hydrodynamic modelling"

_Hydrology and Earth System Sciences, 2018_

## Referee Comment (RC1) · Anonymous Referee #1 · 25 Oct 2018

The authors are commended for trying to use hydrodynamic models to forecast impacts of climate change on drinking water quality. The paper has several serious issues as shown below in the comments:

1. P. 1 Line 15 – the most important point is not a model result but why the model result is occurring. For example, why is there a 2-3 fold increase in E. Coli?

2. Line 64: "When properly calibrated" – what does that mean? What is the proper calibration for hydrodynamic models?

3. Line 71,72: English grammar

4. Section 2.2.1 – If I understood, there was never any calibration of the SWAT model for this watershed – only scaling the hydrologic parameters to the drainage basin areas. This may be OK for a first approximation, but the error doing this must be quantified in order to see the impact on the overall model.

5. Line 138-141: So the overall model was only run for 3 separate years? The detention time of the lake needs to be adequately evaluated – I did not see the volume of the lake. If the lake volume shows that the detention time is over 1 year – then runs must be done for more than just 1 year.

6. Line 169: How was hourly cloud cover measured – this is very unusual!

7. Line 173: This equation 1 is totally unclear – is D the depth of 99 m or variable over the lake? How was this equation used? This does not make any sense – there is important information missing. How was the computation of T from this equation used? It cannot be the temperature model for the lake. This is totally confusing and incorrect.

8. Line 201-203: These equations are incorrectly written.

9. Line 206-207: Ax, Ay, Az are NOT constituent dispersion coefficients.

10. Line 214: Equation is written incorrectly.

11. Line 219: Show the equation of state from Gill.

12. Line 222-225: Upwinding for constituent transport is the worst possible numerical scheme – use a model that has higher order scheme or else show that this scheme was 'fixed' by using an appropriately small grid.

13. Line 229: "square grids of 1 m" – Are you sure? 1m!

14. Line 257: "Accordingly, the model outputs may reflect the actual water temperature". There is nothing presented that gives confidence that the model reflects the actual temperature. Just saying so does not mean that it is true. There needs to be a rigorous analysis of model error – which in this case is too large and shows that the

model is not yet adequately calibrated.

15. Line 261: "the model closely predicted the profiles' – I disagree – the errors for temperature are systematic and large. The model is not adequately calibrated.

16. Line 262-273: What are error bars in measurement of E coli? There was no attempt to quantify the error in the E. Coli and relate that tuo model predictions which are very low. Why would the authors say the model agrees more with 2015 than 2017 for which they have data – that shows something is wrong.

17. Line 275-293 – does the model predict ice cover? If it does, how well does it model ice? The model is not useful unless ice cover is being predicted. There was no mention of ice cover in the paper.

---

## Author Comment (AC1) · 16 Nov 2018

Response to Comment #1. This is very important, and further discussions on the reasons for the predicted changes shall be included in the revised manuscript. For instance, the predicted increases in E. coli in the Lake may be associated with the changes in streamflow and E. coli concentrations in the tributaries of the Lake in the future, as predicted in SWAT. Changes in streamflow may lead to changes in the concentrations of suspended particles and associated faecal indicator organisms that reach the Lake. In addition, variations in the flow regimes affect the mixing conditions in the lake, and this led to more perfect mixing as indicated in the results (Figure 7). Moreover,

the projected changes in air temperature used in the predictions could have played a significant role, since it is a major factor that affects the persistence and regrowth of microorganisms in water.

Response to Comment #2. The statement only reflects some of the challenges encountered in calibrating the model. Apart from data challenges, not all existing hydrodynamic modelling systems fully account for the variety of processes that affect the fate of microorganisms in both sediments and water column. Moreover, we found that certain solution schemes for the governing hydrodynamic equations may have varying accuracy levels when implemented in different water bodies of different dimensions and environmental conditions.

Response to Comment #3. Well noted and would be corrected in the revised manuscript.

Response to Comment #4. 1. Because there was no local flow data, for the catchment of the Lake, hydrological parameter regionalization method was applied to estimate flow in the major streams. The parameters were used to predict potential flow levels in the future using adjusted precipitation and air temperature data. 2. The SWAT calibrations were performed on the major streams, but not on the minor streams since the required spatial data were not available for those small streams. Therefore, flow in these streams were scaled from the major ones. 3. For E. coli calibrations, we used measured local biweekly data from 8 locations including the major streams.

These have been explained in the discussion as part of the limitations of the study. This would be made more succinct in the revised manuscript.

Response to Comment #5. This is a very insightful comment and might be considered in the revision. The volume of the lake is approximately 250, 000 m3. The retention time of the Lake is indeed more than 1 year. However, the model calibration was restricted one year due to the following reasons: - Simulation stability when running long simulations: Smaller timesteps equals higher stability but increases the time to

run the model. To keep the model runs to a realistic time and to keep the stability of the model in check, a model run of one year was found to be the most realistic solution. - Lack of computer processing capacity to run longer simulations in a reasonable time. If the work is to be improved/continued, this is an important part to think of. - We focused on finding the effect of the yearly circulation periods, which happens during spring and autumn, on the climate projections. Without accounting for the retention time, we still see effects of the future climate projections on the lake compared to the current climate.

Response to Comment #6. The collected data was measured every three hours during daytime from 7 a.m. to 7 p.m. Since the model required all meteorological data to have the same time step, the cloud cover data was interpolated to get hourly values.

Response to Comment #7. Eqn. 1 is a simple water temperature model, which describes the rate of change of response temperature. In the eqn., the water temperature is assumed to respond only to surface heat exchange, which is calculated from the other weather parameters as shown in eqn. 2.

D in eqn. 1 is the mean temperature of the water column, and it is variable. The total depth of the water column is rather 99 m.

Response to Comment #8. That is true, some corrections are needed. For instance, the kinematic viscosity terms are wrongly placed, and the signs of the lateral momentum terms ($M_x$ and $SM_y$) need to be reversed.

Response to Comment #9. That is true. They refer to the horizontal and vertical turbulent kinematic viscosities

Response to Comment #10. Well noted and will be corrected accordingly.

Response to Comment #11. The equation would be added in the revision.

Response to Comment #12. We agree with the reviewer on the limitations of the upwind scheme used in this study. The model was run with other higher order schemes (QUICKEST and QUICKEST with ULTIMATE). However, these schemes yielded very

unrealistic results after repeated iterations. We have previously calibrated another model for a smaller, more shallow lake with a large catchment area. In that case, the higher order schemes; QUICKEST and QUICKEST with ULTIMATE had better fits to the calibration data than the upwind first order.

The Lake (Brusdalsvatnet) for which the model was calibrated in this work is a large, deep lake with a small catchment area, the QUICKEST and QUICKEST with ULTIMATE schemes produced poorer fits than the upwind first order scheme. During the calibration with the higher order schemes, outputs of the temperature profiles and the raw water intake point did not fit the measured data even when model parameters were stretched to their limits (minimum to maximum) to make this happen. The closest fits to the observed data as presented in the results were obtained by the Upwind First Order scheme.

Nonetheless, it may however be possible to achieve better fits with the higher order schemes in other models/software.

Response to Comment #13. Indeed, the grids were not 1m in dimension. Only the vertical dimensions were 1 m. The sentence shall be revised accordingly. The horizontal grid is approximately 100 m x 100 m while the vertical layers are every 1 m.

Response to Comment #14. The calibration temperature data (temporal data), which was compared with the model outputs were measured at the outlet of the treated water reservoir in the water treatment plant, rather than at the raw water intake depth. This means that there might be some difference in temperature from the data and the actual temperature at the intake depth. The retention time, from when the water is pumped from intake level to the measurement of temperature at the outlet of the treated water reservoir, is calculated to be approximately 8 hours. While this may not be the only reason for the relatively large difference between the model outputs and the observations, considerable changes in the water temperature could occur from pumping through treatment steps and in the storage reservoir for the treated water. Therefore,

the temporal measurements of water temperature might not be viable calibration data for the model.

This was the main reason for the statement as quoted by the reviewer. We agree that this must be made clearer in the discussion and would be done accordingly in the revision.

Alternatively, we might have to remove the water intake calibration data from the paper so that the calibration data for the temperature model will comprise only the manually measured profile data.

Response to Comment #15. With the input data that was available for the area and the capabilities of the modelling software, we believe this was the best fit we could achieve after many model runs, setting each parameter as best as possible.

Response to Comment #16. The E. coli data as taken from the water treatment plant were composed of detects (count) or non-detects (zeros). Therefore, it was not possible to indicate the measurement errors in the plots.

The 2015 data was added to the plot because as shown in the figure, only two detections of E. coli were recorded in the raw water in 2017. The measurement method has an accuracy of detectable E. coli per 100ml of sample. For example, there could have been up to 9 E.coli colonies per 1L without the test finding any detects. Also, the treatment plant produces 0.37 m3/s while only 100ml of water is tested per week. This is therefore not a well-represented amount of sample.

The 2015 measurements however reflect a typical pattern of observation in the lake, which was predicted by the model. The point was not necessarily to replicate the 2015 data, but to show correlation between the model outputs and the pattern of the 2015 measurements, since more detects were shown in 2015 than in the 2017 measurements.

Response to Comment #17. The model predicts ice cover. However, data on ice cover

was not available for comparison, thus, the output was not presented in this work.

The modelled Lake (Brusdalsvatnet) is located in a mild coastal climate, where ice-forming is not as common and long-lasting as in most other lakes in Norway, therefore, the Lake is never fully covered by ice. The mild climate prevents long periods of cold weather, which is needed to form a stable long-lasting ice cover. Therefore, ice normally only covers smaller portions of the Lake and for short time periods. The model result regarding ice cover is shown in the figures attached.

The figures below show ice cover on the Lake surface predicted by the model in 2017. The ice started appearing on 9th February and ended after 16th February. Ice was formed again on the 7th of March and ended after March 9th.
* * *
**Ice thickness mm  02/09/2017 00:00**

**Ice thickness mm  02/10/2017 00:00**

**Ice thickness mm  02/11/2017 00:00**

**Ice thickness mm  02/12/2017 00:00**

**Ice thickness mm  02/13/2017 00:00**

Fig. 1. Ice cover on lake surface from 9th February to 13th February and 14th to 16th February 2017.

Ice thickness mm   02/14/2017 00:00

Ice thickness mm   02/15/2017 00:00

Ice thickness mm   02/16/2017 00:00

Fig. 2. Ice cover on lake surface from 14th to 16th February 2017.

**Fig. 2.** Ice cover from model output 2

**Ice thickness mm   03/07/2017 00:00**

**Ice thickness mm   03/08/2017 00:00**

**Ice thickness mm   03/09/2017 00:00**

Fig. 3. Ice cover on lake surface from 7th March to 9th March 2017.

**Fig. 3.** Ice cover from model output 3

---

## Referee Comment (RC2) · Anonymous Referee #2 · 22 Nov 2018

The authors used a hydrodynamic and water quality modelling approach to predict the potential impact of climate change projections on water temperature and E. coli concentrations in a raw water source in a lake in Norway. Although the forecast was good for the years 2045 and 2075, compared with the year 2017, there are some revisions that should be made.

The major concerns are with the validation/calibration of the model. For instance:

1. Line 233-234, How was the water sampling performed for E. coli counts? How could the authors simulate the E. coli profile just using the surface and raw water intake point? Did the authors perform any sampling at different depths for E. coli counting,

besides surface and intake point? If the authors measured temperature profiles why did they not performed the same approach for the E. coli counting? I suggest a sentence explaining how it was done because it is not very clear how the model was calibrated only with these two sampling points.

2. What was the technique to measure the E. coli counting, was it membrane filtration? If so, how could the authors get concentrations of 45524 CFU/100 ml, it had to be a huge dilution. Although this was not part of the objective of the experiment, these questions, in my opinion, should be considered for a better understanding and calibration of the model.

Other minor revisions should be considered:

Abstract, line 17, the sentence "The results is expected to…" should be corrected for "The results are expected to…".

Section 2.1 – In the description of the lake, I would suggest including a sentence saying the classification of the lake concerning the type of mixing. Is it a dimictic lake?

Section 2.2.2 Microbial discharge into the lake, line 146, the method and units used to determine the E. coli concentration in water samples should be referred to in the text in this section.

Line 157, at the end of the sentence I would suggest including the reference of Table 1.

Line 215, please, explain better the terms of this equation.

Line 158, units of E. coli concentrations is missing

Line 277, Figure 4 shows the distribution of temperature and concentration of E. coli in the Lake in 2017 during the four major seasons or cross-sections from the model output? Do the numbers 15 after the month corresponds to the year of 2015? Shouldn't it be the year 2017? I suggest a clarification of the legend and figure.

One thing that is not very clear is that, although the authors say that the major streams are the key source of E. coli load on the Lake (line 331-332), "Under the current climate forecast for the catchment area of the Lake, the concentrations of E. coli in the Lake. . . is expected to marginally increase by 2075" (line 395-397) but table 2 shows that average concentrations of E. coli in the tributaries tend to decrease by the year 2075. Also, in table 2, the Arsetelva and Vasstrandelva streams, although they are the "key sources" they exhibit the lower average concentrations. So maybe it should be clear that, perhaps, the "key source" of bacterial contaminations are not the major streams but the populated areas surrounding the north-western part of the Lake.

---

## Author Comment (AC2) · 17 Dec 2018

Comment #1. Line 233-234, How was the water sampling performed for E. coli counts? How could the authors simulate the E. coli profile just using the surface and raw water intake point? Did the authors perform any sampling at different depths for E. coli counting, C1. Besides surface and intake point? If the authors measured temperature profiles why did they not performed the same approach for the E. coli counting? I suggest a sentence explaining how it was done because it is not very clear how the model was calibrated only with these two sampling points.

Response to Comment # 1.
The sampling was performed in the streams on biweekly basis, and the analysis was done using membrane filtration. Delusions of up to 10-2 were used for samples that showed relatively higher E. coli concentrations.

The E. coli data at the water intake point of the lake were obtained from the water treatment plant, who also enumerate the bacteria using membrane filtration. Analysis of samples at different depths were not performed. This could be a very good option for future investigations.

The temperature profiles were also measured by the water treatment plant managers.

The model was calibrated by manually setting parameters and running model simulations until the best fit was found. The model calibration was fitted to measured temperature data were both the profiles and intake data were used. The E. coli data measured at the streams were used as input data in the temperature calibrated model. The e. coli data in itself was not calibrated. But t o see how the current peak of E.coli could be in the future, scaling of the input data was done to match the measured peak at the intake depth before running the future scenarios (2045 and 2075).

Comment #2. What was the technique to measure the E. coli counting, was it membrane filtration? If so, how could the authors get concentrations of 45524 CFU/100 ml, it had to be a huge dilution. Although this was not part of the objective of the experiment, these questions, in my opinion, should be considered for a better understanding and calibration of the model.

Response to Comment # 2.

See response to comment # 1 above. The 45524 CFU/100ml is after scaling, see response to comment # 1.

Comment #3. Abstract, line 17, the sentence "The results is expected to..." should be corrected for "The results are expected to...".

Response to Comment # 3.

The comment is well noted and would be corrected in the revision.

Comment #4. Section2.1–In the description of the lake, I would suggest including a sentence saying the classification of the lake concerning the type of mixing. Is it a dimictic lake?

Response to Comment # 4.

The lake is mainly dimictic. Thermoclines occur in the summer, while mixing occurs in spring and autumn. The condition in the winter however depends on the water temperature.

Moreover, the vast cover of mountains that surround the lake reduces the effect of wind on the mixing conditions. Thus, mixing is not complete throughout the lake.

Comment #5. Section 2.2.2 Microbial discharge into the lake, line 146, the method and units used to determine the E. coli concentration in water samples should be referred to in the text in this section.

Response to Comment # 5.

The comment is well noted and the method used in determining the E. coli concentrations would be explained and referred to accordingly.

Comment #6. Line 157, at the end of the sentence I would suggest including the reference of Table 1.

Response to Comment # 6.

Well noted and would be done accordingly

Comment #7. Line 215, please, explain better the terms of this equation.

Response to Comment # 7.

Further clarifications about the terms in the equation will be given in the revision.
Comment #8. Line 158, units of E. coli concentrations is missing.

Response to Comment # 8.

Well noted. The units of E. coli shall be added accordingly.

Comment #9. Line 277, Figure 4 shows the distribution of temperature and concentration of E. coli in the Lake in 2017 during the four major seasons or cross-sections from the model output? Do the numbers 15 after the month corresponds to the year of 2015? Shouldn't it be the year 2017? I suggest a clarifi̧cation of the legend and fi̧gure.

One thing that is not very clear is that, although the authors say that the major streams are the key source of E.coli load on the Lake(line331-332), "Under the current climate forecast for the catchment area of the Lake, the concentrations of E. coli in the Lake... is expected to marginally increase by 2075" (line 395-397) but table 2 shows that average concentrations of E. coli in the tributaries tend to decrease by the year 2075. Also, in table 2, the Arsetelva and Vasstrandelva streams, although they are the "key sources" they exhibit the lower average concentrations. So maybe it should be clear that, perhaps, the "key source" of bacterial contaminations are not the major streams but the populated areas surrounding the north-western part of the Lake.

Response to Comment # 9.

The distributions shown in Figure 4 were taken on the 15th of the months (march, July, November, and January), which were respectively assumed to as the middle of each of the four seasons; spring, summer, autumn and winter. The year was however 2017.

The values in Table 2 show the average concentrations of E. coli observed in the streams in 2017, compared with the predictions for 2045 and 2075. However, the statement as quoted by the reviewer refers to the predicted concentrations at the raw water intake point of the lake (35 m below surface).

The two streams; Arsetelva and Vasstrandelva are two of the major streams. However,

the other two major streams; Slettebakk and Brusdalen were identified as the "most important" sources of E. coli.

We agree with the reviewer that other potential sources including the populated areas could be more important sources, although no discharging streams are located in these areas. Therefore, further explanation shall be given in the revision to clarify this.

The decrease in tributary E.coli concentrations is partly because of higher flows in the tributaries. The overall microbial impact of the tributaries is increasing.

In our model two of the streams, Slettebakk and Brusdalen has the highest concentrations of E.coli. However, the two streams, Vasstrandelva and Arsetelva has the highest flow. The overall microbial impact of a stream is a combination of the flow and the concentration of E.coli in the flow. Line 331-332 talks about all four streams.

Because of the scaling of the E.coli input data to fit the peak measured level at the intake point, we agree that other sources are likely to be more important.

———————————————